# *son* is necessary for proper vertebrate blood development

**Rebecca L. Belmonte**[1], **Isabella L. Engbretson**[1], **Jung-Hyun Kim**[2], **Illiana Cajias**[1], **Eun-Young Erin Ahn**[3,4], **David L. Stachura**[1] *

1 Department of Biological Sciences, California State University Chico, Chico, California, United States of America, 2 Mitchell Cancer Institute, University of South Alabama, Mobile, Alabama, United States of America, 3 Department of Pathology, University of Alabama at Birmingham, Birmingham, Alabama, United States of America, 4 O'Neal Comprehensive Cancer Center, University of Alabama at Birmingham, Birmingham, Alabama, United States of America

* dstachura@csuchico.edu

**Data Availability Statement:** All relevant data are within the manuscript and its Supporting information files.

**Funding:** This research was supported by the CSU Program for Education and Research in

## Abstract

The gene *SON* is on human chromosome 21 (21q22.11) and is thought to be associated with hematopoietic disorders that accompany Down syndrome. Additionally, *SON* is an RNA splicing factor that plays a role in the transcription of leukemia-associated genes. Previously, we showed that mutations in *SON* cause malformations in human and zebrafish spines and brains during early embryonic development. To examine the role of *SON* in normal hematopoiesis, we reduced expression of the zebrafish homolog of *SON* in zebrafish at the single-cell developmental stage with specific morpholinos. In addition to the brain and spinal malformations we also observed abnormal blood cell levels upon *son* knockdown. We then investigated how blood production was altered when levels of *son* were reduced. Decreased levels of *son* resulted in lower amounts of red blood cells when visualized with *lcr*:GFP transgenic fish. There were also reduced thrombocytes seen with *cd41*:GFP fish, and myeloid cells when *mpx*:GFP fish were examined. We also observed a significant decrease in the quantity of T cells, visualized with *lck*:GFP fish. However, when we examined their hematopoietic stem and progenitor cells (HSPCs), we saw no difference in colony-forming capability. These studies indicate that *son* is essential for the proper differentiation of the innate and adaptive immune system, and further investigation determining the molecular pathways involved during blood development should elucidate important information about vertebrate HSPC generation, proliferation, and differentiation.

## Introduction

Blood development (hematopoiesis) is the process by which blood cells acquire specific characteristics through carefully regulated gene expression [1]. All blood cells arise from hematopoietic stem cells (HSCs), which are adult stem cells capable of self-renewal and differentiation into more developmentally restricted blood progenitor cells [2–4]. HSCs differentiate into either a common myeloid progenitor (CMP) or a common lymphoid progenitor (CLP). The CMP can produce all erythromyeloid cells, which are red blood cells, thrombocytes,

Biotechnology (CSUPERB) (New Investigator Awards to D.L.S.), NSF MRI (proposal 1626406), a California State University Chico Internal Research Grant (to D.L.S.), Student Research and Creativity Awards from California State University, Chico (to R.B.) and the NIH (R01CA190688 to E.-Y.E.A. and R15DK114732-01 to D.L.S.). The funders had no role in study design, data collection and analysis, decision to publish, or preparation of the manuscript.

**Competing interests:** D.L.S. is a scientific consultant and has received compensation from Finless Foods, Inc. and Xytogen Biotech, Inc. This does not alter our adherence to PLOS ONE policies on sharing data and materials. All other authors declare no competing interests.

granulocytes, and monocytes [5, 6]. The CLP can produce lymphoid cells, such as B cells, T cells, and NK cells [7, 8]. In essence, red blood cells (which carry oxygen to tissues), granulocytes (which fight infections), thrombocytes (which prevent bleeding), and lymphocytes (which confer adaptive immunity) are very different mature blood cells with diverse functions that all originated from HSCs. Many molecular factors affect the differentiation of blood cells by changing the gene expression of progenitor cells, and this control is essential- if the wrong numbers or wrong types of mature blood cells are produced it can have serious clinical effects on an individual. Understanding the molecular control of these processes is essential for understanding blood development and treating human diseases. With the advent of gene therapy, discovering mutant genes associated with abnormal blood development will also allow their correction, providing new treatments for blood diseases.

Our gene of interest, *SON*, is located on human chromosome 21 and may play a role in regulation of gene expression associated with the Down syndrome phenotype. Down syndrome patients with trisomy 21 have increased levels of transient myeloproliferative disease (TMD), acute megakaryocytic leukemia (AMKL), and acute lymphoid leukemia (ALL) [9]. *SON* codes for a protein that forms an RNA-splicing molecule important for processing other genes during transcription, especially several involved in cell-cycle progression [10]. *SON* was also described as a negative regulatory element binding protein (NREBP) due to its transcriptional repression of human hepatitis B virus genes [11] and regulates the transcriptional initiation of leukemia-associated genes in human and mouse cells [12]. *SON* was later classified as a regulator of pre-mRNA splicing and a nuclear speckle protein [13]. In 2011, it was further shown that *SON* functioned in pre-mRNA splicing [10, 14]. *SON* also has a role in regulating the pluripotency of human embryonic stem cells [15] and is highly expressed in HSCs and downregulated during hematopoietic differentiation [16]. Previous work we performed indicates that mutations in *SON* are linked to spinal and brain malformations in human patients [17], and these patients also have hematologic and kidney abnormalities [18, 19]. Due to the ability of *SON* to regulate multiple genes, mutations seen in human patients, and its expression in HSCs, we hypothesized that *SON* played a role in regulating blood development and formation of the immune system.

To understand the effects of *SON* in blood development, zebrafish (*Danio rerio*) were utilized as a model system. These experiments are not possible to conduct on humans, and zebrafish have many advantages over other animal models. Zebrafish have a well-conserved genome compared to humans, and they undergo rapid development [20, 21]. It is difficult to study blood development in mammals because they develop *in utero*, but zebrafish develop externally and are optically transparent at the larval stage, aiding in their visualization during embryonic development. Overall, zebrafish have been widely established as a model organism to study hematopoiesis [22, 23]. To study how *SON* affects blood development, we utilized transgenic zebrafish that have blood-specific gene promoters driving green fluorescent protein expression in specific blood cell types. These fluorescently labelled cells allow for easy *in vivo* visualization of blood in developing embryos and it allows the use of flow cytometry to interrogate and quantify the numbers of these cells present in an animal [20, 24]. To understand how each branch of hematopoiesis is affected, we used zebrafish that have fluorescently labeled red blood cells, thrombocytes, myeloid cells, and T cells.

For this investigation, we utilized previously validated morpholinos (MOs) [17, 19] to study the role of SON in blood development. While there is not a zebrafish gene annotated in the genome named "*SON,*" the well-conserved ortholog to human *SON* is on an unplaced scaffold of the zebrafish genome (NCBI Gene: LOC565999) identified as CABZ01113192.1 (www. ensembl.org, assembly GRCz11). Although the protein encoded by CABZ01113192.1 is much smaller than the human SON protein, the domains critical for SON's function (*i.e.* serine/

arginine-rich domain, G-patch, and double-stranded RNA-binding motif) are well conserved between these two species. As these genes are orthologous (see [17] for alignments between zebrafish and human orthologs), throughout the paper we refer to CABZ01113192.1 as *son*. MOs we used are specific antisense oligonucleotide that bind only to *son* mRNA; this binding alters its splicing, reducing the levels of normal Son protein in the developing fish [17, 19, 25]. Zebrafish are excellent model organisms to use with MOs, since they have been established as an inexpensive, effective method of knocking down levels of specific genes [26]. MOs allow researchers to reduce gene products versus causing their complete ablation, which can sometimes be lethal, especially with genes that may have pleiotropic effects. MOs also do not cause genetic compensation, an issue recently discovered with CRISPR-mediated gene knockout [27].

Overall, reducing *son* and investigating blood and immune cells indicates that this gene plays an important role in early hematopoietic development. After injecting single-cell embryos with *son* MO, we saw a significant decrease in the amount of red blood cells, thrombocytes, myeloid cells, and T cells. However, we did not see a difference in the quantity of erythromyeloid hematopoietic stem and progenitor cells (HSPCs), indicating that *son* is involved in proper blood maturation. While we don't yet understand the specific molecular mechanism by which this happens, this study offers insight into an important gene regulating normal blood cell development that is altered in human diseases. Additionally, it will offer new avenues for treatment of hematologic disorders associated with dysregulated SON expression.

## Materials and methods

### Zebrafish husbandry and care

Zebrafish were mated, staged, and raised as described [28] and maintained in accordance with California State University (CSU), Chico Institutional Animal Care and Use Committee (IACUC) guidelines. All procedures were approved by the CSUC IACUC before being performed. Personnel were trained in animal care by taking the online Citi Program training course entitled "Working With Zebrafish (*Danio rerio*) in Research Settings" (https://www.citiprogram.org). Wildtype (wt) and the transgenic zebrafish lines *lcr*:GFP [29], *mpx*:EGFP [30], *cd41*:GFP (also known as *itga2b*:GFP) [31], *lck*:EGFP [32], and *fli1a*:EGFP [33] were used for these studies. Zebrafish were housed in a 700L recirculating zebrafish aquarium system (Aquatic Enterprises, Seattle, WA) regulated by a Profilux 3 Outdoor module that regulated salinity, pH, and temperature (GHL International, Kaiserslautern, Germany) 24-hours-a day. The facility was illuminated on a 14-hour light/ 10-hour dark cycle. Zebrafish were fed once a day with hatched brine shrimp (Brine Shrimp Direct, Ogden, UT) and once a day with Gemma micro 300 (Skretting, Westbrook, ME). After experiments were performed all animals were returned to the aquarium system to be used for further research.

### MO injection

To examine the specific function of *son*, 1nL of *son* MO (Gene Tool, LLC, Philomath, OR) was injected at 6.25μM into single-cell zebrafish embryos, resulting in a total injection of 6.25ng of the MO into each individual. This amount was used based on previous studies [17, 19]. The *son* MO sequence is 5′-TGGTCCTGGATATAACAGACAGATT-3′ [17]. Control MOs were also utilized (5′-CCTCTTACCTCAGTTACAATTTATA-3′) at the same concentration.

### Rescue assays

*son* cDNA was subcloned into pcDNA3.1$^{+/-}$ and linearized with Sca1. *son* mRNA was generated using a mMessage T7 kit (Ambion, Austin, TX). 1nL of *son* MO (at 6.25μM) and 0.5nL of

*son* mRNA (at 1μg/μL) was injected into single-cell zebrafish embryos, resulting in a total injection of 6.25ng of the MO and 500ng mRNA into each individual. *son* mRNA was visualized after being electrophoresed on a 1% w/v agaose gel with TAE and 1% bleach added (S1 Fig).

## Microscopy

All observations were made with a Leica M165C microscope and pictures were taken with a Leica DFC295 camera. *lcr*:GFP, *mpx*:GFP, and *fli1*:GFP embryos were sorted at 24 hours post fertilization (hpf) and observed and imaged at 48hpf. Images of *mpx*:GFP embryos were taken at 48hpf and the number of fluorescent cells per embryo were enumerated. To perform this, embryo images were taken at the same exposure and every *mpx*:GFP$^+$ cell was manually counted. To prevent bias, the images were coded and students counted the *mpx*:GFP$^+$ blinded from the experimental variable. *cd41*:GFP embryos were sorted into phenotypic groups, counted, and imaged at 72hpf in a similar manner. *lck*:GFP embryos were observed at 5 days post fertilization (dpf) and images were taken of each embryo. For thymus quantitation, every image was taken at 80x and the focal plane was adjusted to best capture the whole thymus. The exposure was kept constant for all images to reduce variation between images. Images of *lck*:GFP embryos were analyzed with ImageJ (https://imagej.nih.gov/ij/) to determine the pixel density of each thymus. Briefly, a bounding box was drawn around the thymus, and ImageJ calculated the numbers of pixels present. To correct for background fluorescence, each image had an equal sized box drawn outside of the embryo's image in a non-fluorescent region. The pixel density of this box was subtracted from the pixel density of the thymus to normalize all the images. To prevent bias, the images were coded and students analyzed the pixel density blinded from the experimental variable.

## Methylcellulose assay

To determine the effect of reduced *son* expression on erythromyeloid HSPC colony formation, clonal methylcellulose assays were performed [24, 34–38]. Briefly, five embryos were selected at random from uninjected animals, digested, and plated in supportive media. Five animals were also randomly selected from the *son* MO-injected group and plated at the same time to compare the colony growth seen in uninjected versus MO-injected samples [38, 39]. The digested embryos were grown in methylcellulose media with carp serum, Gcsf, and Epo added to selectively stimulate myeloid and erythroid differentiation from HSPCs. Samples were incubated at 5% $CO_2$ at 32°C for 7 days. Each sample was observed at 40x with an Olympus IX53 inverted microscope (Olympus Life Science, Waltham, MA) and colonies were counted from each sample. Each point shown on the graph is a biological replicate of five randomly selected embryos from uninjected versus MO-injected categories.

## Reverse transcriptase PCR (RT-PCR)

RNA was extracted with the RNeasy Plus Mini Kit (Qiagen, Hilden, Germany) from uninjected embryos and those injected with *son* MO at 24hpf, 48hpf, and 72hpf. iScript (Bio-Rad, Hercules, CA) was used to generate cDNA from each sample. Each cDNA sample was made by pooling ten random whole embryos per condition. PCR was performed with JumpStart TAQ polymerase ready mix (Invitrogen, Carlsbad, CA) with the following primers for *son*: FWD: 5'-ATGGAGAAAATCCAACTGTG-3' and REV: 5'-GACCTTAAAGAGGAAGTTC-3'. *ef1α* [40] was used as a reference gene.

## Quantitative reverse transcriptase PCR (qRT-PCR)

RNA was extracted with the RNeasy Plus Mini Kit (Qiagen, Hilden, Germany) from unin-jected embryos and those injected with *son* MO and those injected with *son* MO and *son* mRNA at 48hpf, 72hpf, and 5dpf. iScript (Bio-Rad, Hercules, CA) was used to generate cDNA from each sample. Each cDNA sample was made by pooling ten random whole embryos per condition. Samples were then run on an Eppendorf realplex2 (Eppendorf, Hamburg, Germany) with primers for *ef1α* [40], *band3* (also called *slc4a1a)* [41], *cmpl* [40], *hbaa1* (FWD: 5'-GGACAAGGCTGTTGTTAAGG-3', REV 5'-AGACCAGTGAGAGAAGTAG-3'), *cd41* [40], *mpx* [42], *csf3r* [42], and *lck* [43]). Data were analyzed for relative expression change with *ef1α* [40] as the reference gene. ΔΔCt was calculated by comparing the expression of the injected embryos to control embryos and to the reference gene, *ef1α*.

## Flow cytometry

To enumerate the percentage of RBCs in an embryo, we used transgenic *lcr*:GFP [29] zebrafish in combination with flow cytometry. 48 hpf transgenic embryos were grouped in samples of ten and washed 3x with Dulbecco's phosphate-buffered saline (DPBS) containing $Ca^{2+}$ and $Mg^{2+}$. After the last wash, 500 μL of DPBS and 5 μL of 5 mg/mL (26U/mL) Liberase TM (Roche, Upper Bavaria, Germany) were added. Samples were incubated at 37˚C on a horizon-tal orbital shaker at 180 rpm for 60 mins. Samples were than triturated with a P-1000 to ensure proper dissociation and transferred to a 5 mL polystyrene round bottom tube with cell strainer cap. Samples were strained, centrifuged, and resuspended in 100 μL of DPBS and 1 μL of SytoxRed (ThermoFisher Scientific, Waltham, MA) was added to each sample to label dead cells. Samples were run through a BD FACSAria Fusion flow cytometer (BD Biosciences, San Jose, CA) and enumerated. Data were analyzed using FloJo software (FloJo LLC, Ashland, Oregon) to quantitate total percentage of positive fluorescent cells.

## Statistical methods

Statistical analyses were performed in Microsoft Excel. To discern statistical difference, data were analyzed using an unpaired two-tailed Student's T test assuming unequal variance. All raw data from these studies are supplied in S1 Data.

## Results

To elucidate the role of *son* in blood development, we used a *son*-specific MO [17, 19] to reduce protein levels. Injection of the MO at the single-cell stage of development and examina-tion of *son* mRNA levels at 24, 48, and 72 hpf indicated a reduction in full-length *son*, likely caused by mis-splicing and nonsense-mediated decay (S2 Fig). These injections allowed us to then observe phenotypic changes caused by a reduction of *son* levels. After seeing no blood-specific phenotypic differences between embryos injected with the control MO and uninjected embryos (data not shown, [17, 38, 39], S3 and S4 Figs), we used uninjected embryos as the con-trol for the remaining experiments. We first performed experiments in a transgenic zebrafish line that has the alpha globin locus control region (lcr) driving GFP expression; these *lcr*:GFP transgenic zebrafish only express GFP in red blood cells (RBCs) [29]. Injection of *son* MO into embryos caused several phenotypes; the majority of injected embryos either had no circulating RBCs or reduced levels of visible RBCs present at 48hpf (Fig 1A–1E). About 10% of injected fish had no discernible RBCs at all, while less than 10% were phenotypically normal (Fig 1A–1E). To confirm these studies, we quantitated RBC numbers in *son* morphants with flow cytometry. Embryos injected with *son* MOs had reduced levels of RBCs when compared to

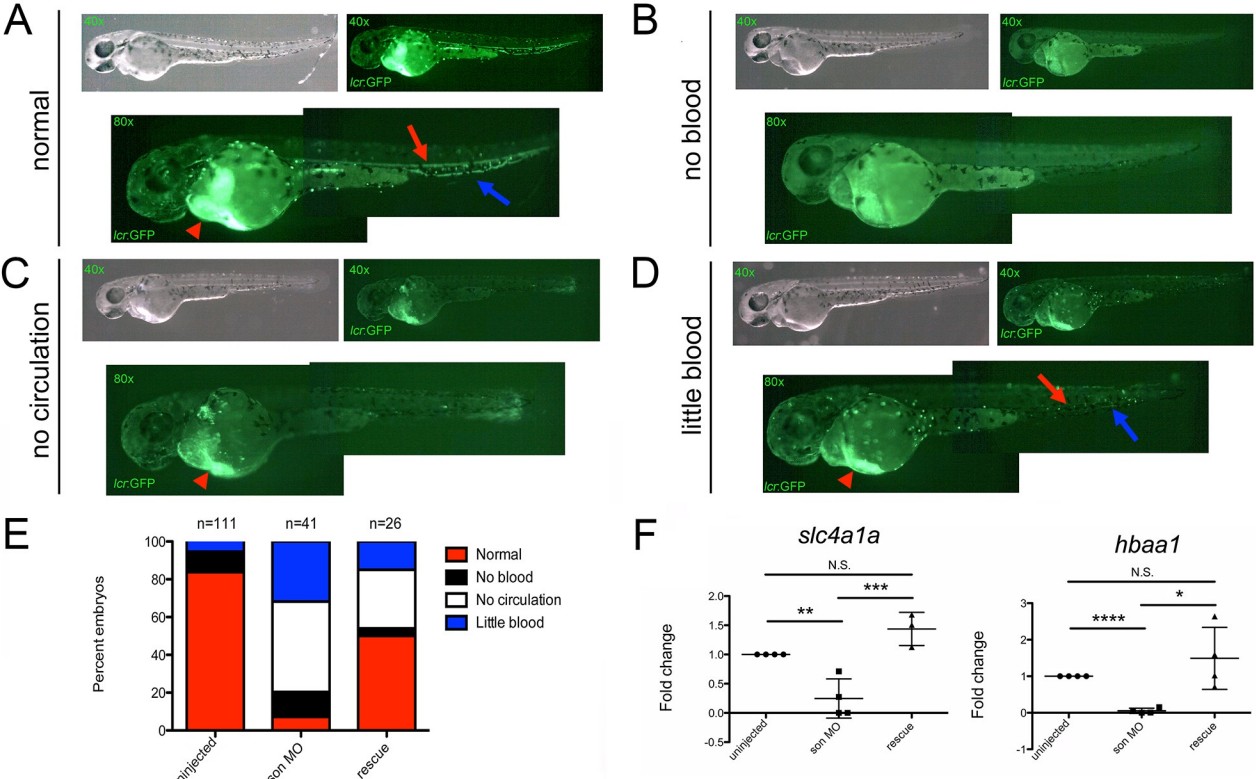

**Fig 1. Reduction in *son* results in impaired erythropoiesis.** Representative images of 48hpf *lcr*:GFP embryos injected with *son* MO at the one-cell-stage of development show four common phenotypic categories associated with *son* MO injection. Category titles are listed to the left of each image. Brightfield images are shown in top left (40x), and fluorescent images are shown in top right (40x). Zoomed in fluorescent images are shown below (80x); each green dot in the animal is a labelled RBC. Fluorescent "trails" of green are cells moving in the vasculature. Red arrows indicate RBCs flowing though the dorsal aorta, and blue arrows indicate RBCs flowing back to the heart through the caudal vein. Red arrowheads indicate large numbers of RBCs pooled in the ducts of Cuvier, located on the yolk ball. (**A**) Representative images of normal RBC numbers and blood flow. (**B**) Representative animals with no RBCs (no blood), (**C**) no RBCs circulating in the animal (no circulation), and (**D**) reduced numbers of RBCs, some of which are circulating (little blood). (**E**) Quantitation of phenotypes shown. Number of embryos analyzed is shown above the chart. (**F**) qRT-PCR for *slc4a1a* (left) and *hbaa1* (right) was performed. Each point represents ten embryos randomly selected from uninjected, MO-injected (*son* MO), or MO-injected with *son* RNA (rescue) conditions that were analyzed by qRT-PCR. Middle lines represent mean and error bars represent SD. * represents p = 0.04, ** represents p = 0.02, *** represents p = 0.004, **** represents p < 0.001, N.S. represents no significance.

embryos injected with control MOs (S3A and S3B Fig), and this was rescued by the addition of *son* mRNA (S3C Fig). Quantitation of these data can be seen in S3D Fig. Additionally, qRT-PCR showed a significant decrease in the expression of *band3 (slc4a1a)* and *hbaa1*, both specific markers of mature erythrocytes, when *son* MO was injected compared to control embryos (Fig 1F). Importantly, rescuing *son* reduction with the addition of *son* mRNA rescued RBC loss (Fig 1E) and restored the expression of *band3* and *hbaa1* (Fig 1F). Together, these data indicate that decreased *son* impairs successful erythropoiesis.

We next examined thrombocytes, cells responsible for blood clotting, by utilizing *cd41*:GFP transgenic animals that have GFP+ thrombocytes [31] (Fig 2). These embryos showed a significant decrease in *cd41*:GFP+ thrombocytes at 72hpf, with most injected animals showing no thrombocytes at all (Fig 2A–2E). Over 15% of embryos had less thrombocytes than uninjected controls, close to 10% had no thrombocytes in circulation, and very few had normal, circulating cells (Fig 2A–2E). We also performed qRT-PCR on these embryos at 72hpf and observed a significant decrease in the thrombocytic markers *cmpl* and *cd41* in injected embryos when compared to the control embryos (Fig 2F). Again, rescuing the loss of *son* expression with

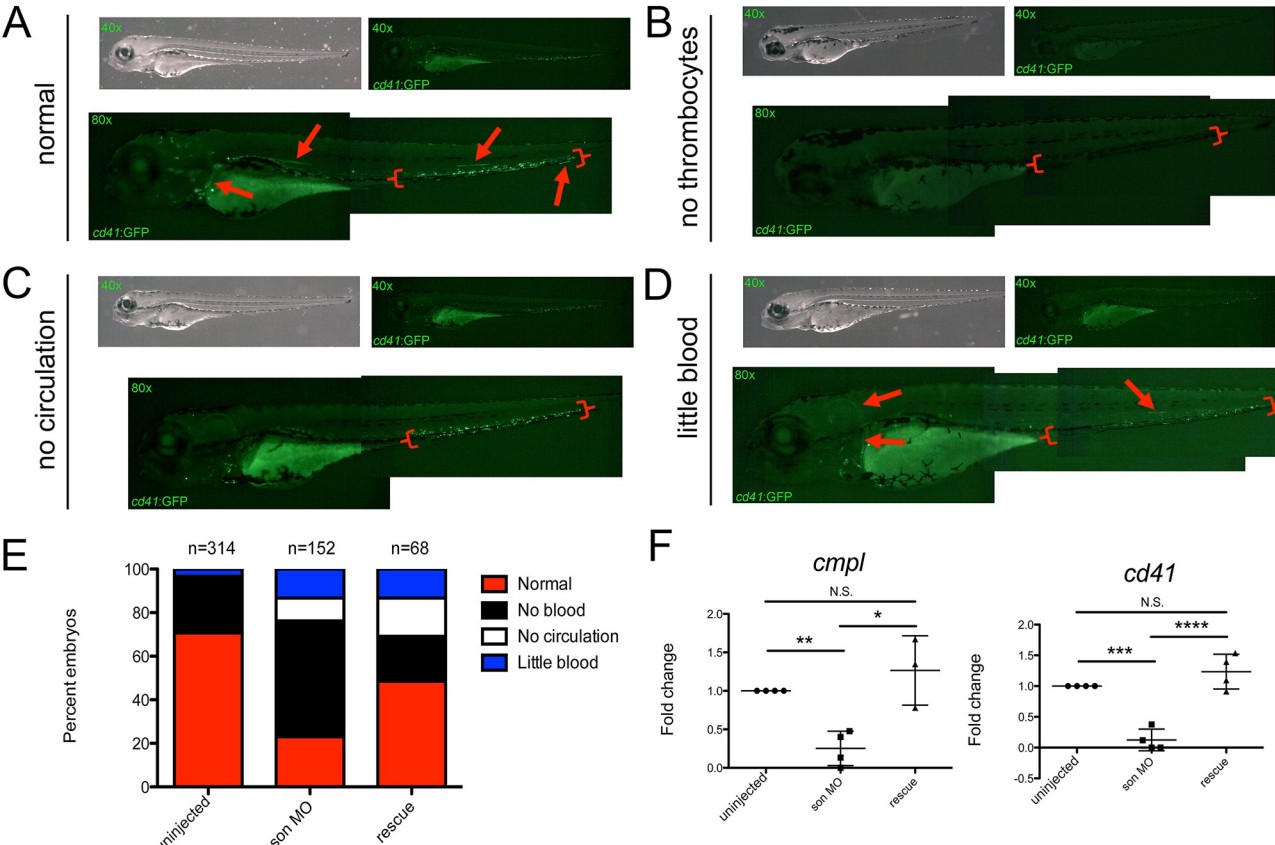

**Fig 2. Reduced *son* expression causes a reduction in thrombocytes.** Representative images of 72hpf *cd41*:GFP embryos injected with *son* MO at the one-cell-stage of development show four common phenotypic categories associated with MO injection. Category titles are listed to the left of each image. Brightfield images are shown in top left (40x), and fluorescent images are shown in top right (40x). Zoomed in fluorescent images are shown below (80x); each green dot in the animal is a labelled thrombocyte. Red arrows indicate fluorescent "trails" made by moving thrombocytes flowing though the vasculature. Red brackets indicate the caudal hematopoietic region where the majority of thrombocytes reside at this time. (**A**) Representative images of normal thrombocyte numbers and circulating thrombocytes. (**B**) Representative animals with no thrombocytes, (**C**) no thrombocytes circulating in the animal (no circulation), and (**D**) reduced numbers of thrombocytes, some of which are circulating (little blood). (**E**) Quantitation of phenotypes shown. Number of embryos analyzed is shown above the chart. (**F**) qRT-PCR for *cmpl* (left) and *cd41* (right) was performed. Each point represents ten embryos randomly selected from uninjected, MO-injected (*son* MO), or MO-injected with *son* RNA (rescue) conditions that were analyzed by qRT-PCR. Middle lines represent mean and error bars represent SD. * represents p = 0.04, ** represents p = 0.006, *** represents p = 0.002, **** represents p = 0.001, N.S. represents no significance.

mRNA restored levels of thrombocytes (Fig 2E) and thrombocytic genes (Fig 2F). Together, these data indicate that *son* is also important for normal thrombocyte production in the embryo.

Next, we investigated the effects of *son* reduction on myeloid cells using *mpx*:GFP embryos, which label neutrophils [30]. 48hpf embryos from the uninjected and *son* MO-injected groups (Fig 3A) were imaged. Then, the number of fluorescent myeloid cells per embryo were enumerated (Fig 3B). The control embryos had a significantly higher number of myeloid cells compared to embryos injected with *son* MO (Fig 3B). We also performed qRT-PCR on these embryos at 48hpf and observed a significant decrease in the myeloid markers *mpx* and *csf3r* in MO-injected embryos when compared to controls (Fig 3C). Once again, *son* mRNA rescued these phenotypic reductions in neutrophils (Fig 3B and 3C). These data indicate that *son* is also crucial for proper myeloid cell development.

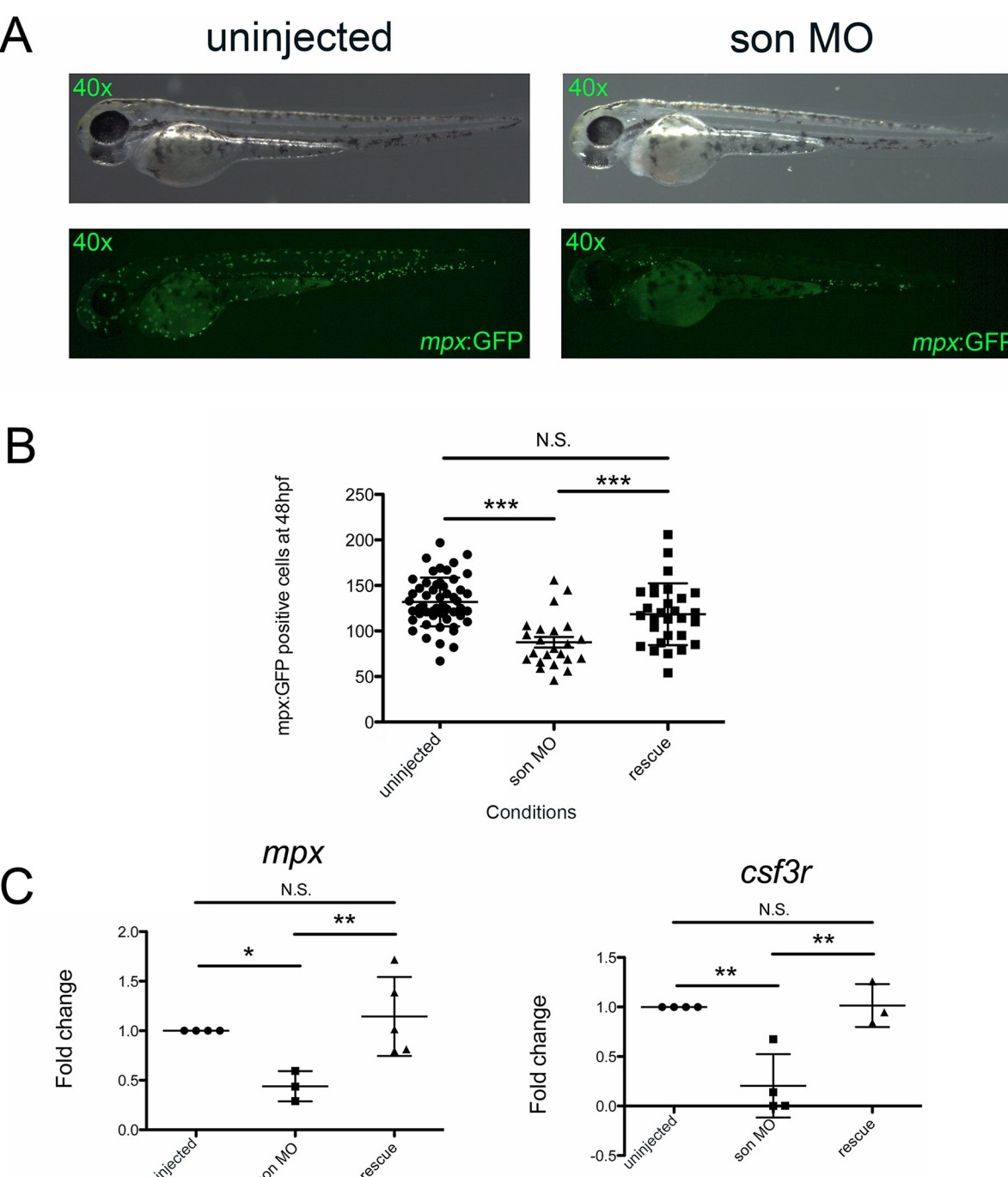

**Fig 3. *son* knockdown causes a reduction in neutrophils.** (**A**) Representative images of *mpx*:GFP embryos at 48hpf that were uninjected (left) or MO-injected (*son* MO, right). Images are taken at 40x, and every green dot is a labelled neutrophil; these cells are in tissues and not circulating. (**B**) Each point represents the number of GFP+ myeloid cells present in an individual uninjected, MO-injected (*son* MO), or MO-injected with *son* RNA (rescue) embryo. Images were taken and GFP$^+$ cells were manually enumerated. Middle lines represent mean and error bars represent SD. (**C**) qRT-PCR for *mpx* (left) and *csf3r* (right) was performed. Each point represents ten embryos randomly selected from uninjected, MO-injected (*son* MO), or MO-injected with *son* RNA (rescue) conditions that were analyzed by qRT-PCR. Middle lines represent mean and error bars represent SD. * represents p = 0.02, ** represents p = 0.01, *** represents p < 0.001, N.S. represents no significance.

Next, we investigated if *son* perturbation negatively affected the number of T cells with *lck*: GFP embryos [32]. Since T cells develop in the thymus, we imaged each fluorescent thymus from individuals at the same exposure time from both the control and *son* MO-injected groups at 5dpf, the time when T cells arise (Fig 4A). We maintained a consistent focal plane to minimize outside effects on the magnitude of fluorescence detected and used ImageJ to calculate the pixel density of each thymus. The control embryos had a higher average thymus pixel density than the *son* MO-injected embryos (Fig 4B). qRT-PCR of these animals at 5dpf for *lck*, a T cell specific marker, also was decreased in *son* morphants (Fig 4C). Again, we observed that addition of *son* mRNA rescued these reductions (Fig 4B and 4C), indicating that *son* is also necessary for proper T cell development.

After observing a decrease in mature blood cell types, we turned our attention to enumerating erythromyeloid HSPCs, which give rise to those blood lineages. We performed these experiments to examine if these key players in hematopoiesis were also reduced or if the loss of blood cells was somehow caused by a failure of HSPC differentiation. To perform these experiments, we used a methylcellulose assay to enumerate the amount of these HSPCs in each embryo at 48hpf [38]. With this assay, either five random uninjected embryos or five random *son* MO-injected embryos were enzymatically digested in separate tubes. The digested embryos were then plated in methylcellulose medium along with hematopoietic-supportive cytokines and growth factors. Under these conditions, HSPCs divide and proliferate, but the medium does not allow the cells to migrate throughout the plate, so each colony that is seen represents one HSPC that was present in the original embryo (Fig 5A). The numbers of colonies were then counted under a microscope to determine the colony forming units (CFUs) present in individual embryos. Interestingly, we saw no significant difference between the number of CFUs from control and *son* MO-injected embryos (Fig 5B). These data indicate that *son* is not crucial for HSPC generation in the 48hpf embryo.

Finally, we wanted to ensure that progenitors were not reduced due to a defect in the vascular system of the developing embryo, as HSCs arise from hemogenic endothelium in the dorsal aorta between 36-52hpf [44, 45]. We also wanted to confirm that the vasculature was not negatively affected in embryos that have severe bending of the spine and other defects associated with son reduction [17–19]. Imaging the vasculature of *son* MO-injected *fli1*:GFP transgenic embryos at 48hpf showed no discernible defects in the dorsal aorta, even if fish were severely bent, a phenotype associated with *son* reduction (S4 Fig). Together, these data point to *son* regulating HSPCs without negatively affecting the formation of mesoderm that gives rise to those progenitors.

## Discussion

We previously showed that zebrafish are an effective model organism for investigating the effects of *son* knockdown with a specific MO and that *son* is necessary for proper brain [17], skeletal [17], and kidney formation [19]. Here we report that *son* is necessary for proper blood formation through an analysis of mature blood cells and embryonic HSPCs. We saw a decrease in the number of RBCs, thrombocytes, myeloid cells, and T cells; however, we did not see a change in the number of erythromyeloid HSPCs. Further studies are warranted to establish the molecular mechanism by which *son* (and its human ortholog *SON)*, directs and manages blood cell differentiation.

In our studies, we show that mature RBCs are reduced in number at 48hpf when measured by fluorescent microscopy, flow cytometry, and qRT-PCR analysis. While we observed many fish with no RBCs and reduced numbers of RBCs, we also saw fish with less RBCs in circulation. It is interesting to note that the morphant fish do not have vascular defects; imaging of

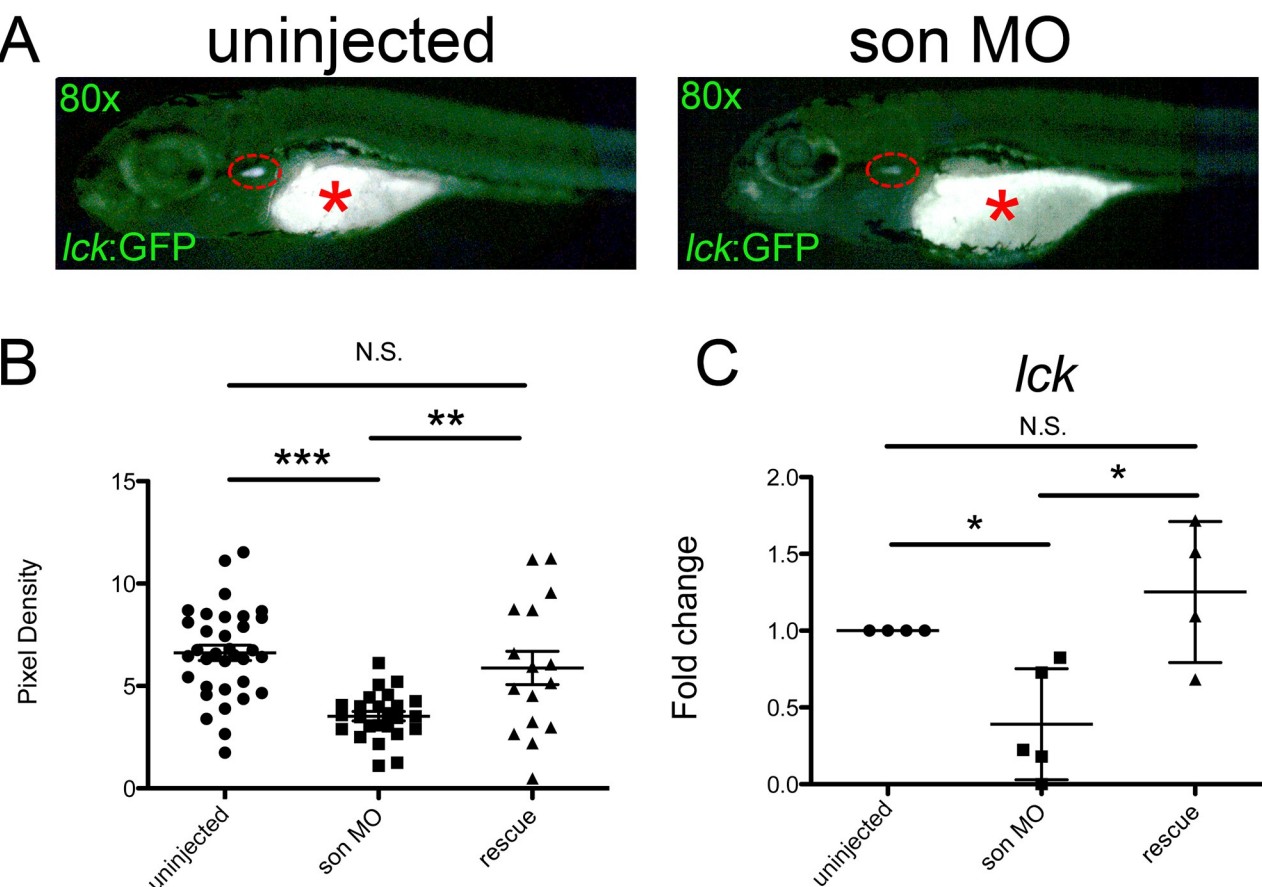

**Fig 4. *son* reduction decreases T cell numbers.** (**A**) Representative images of *lck*:GFP embryos at 5dpf that were uninjected (control; left) or injected with MO at the one-cell-stage of development (*son* MO, right). Images were taken at 80x. Individual GFP⁺ T cells are located in the thymi (red dashed oval); these cells are not circulating. * denotes background fluorescence present in the yolk ball due to refraction of light from lipids present in the yolk. (**B**) Images (like shown in **A**) were subjected to analysis with ImageJ to determine the pixel density of uninjected, MO-injected (*son* MO), or MO-injected with *son* RNA (rescue) thymi. (**C**) qRT-PCR for *lck* was performed. Each point represents ten embryos randomly selected from uninjected, MO-injected (*son* MO), or MO-injected with *son* RNA (rescue) conditions that were analyzed by qRT-PCR. Middle lines represent mean and error bars represent SD. * represents p = 0.02, ** represents p = 0.01, *** represents p < 0.001, N.S. represents no significance.

*fli1*:GFP embryos indicated no issue with vessel integrity or formation, and visible liquid could be seen circulating through the vessels. We also saw a reduction in thrombocytes at 72hpf when measured by fluorescence microscopy and qRT-PCR. Again, in addition to fish with no thrombocytes and reduced numbers of thrombocytes, we observed fish with no thrombocytes in circulation. Examination of non-circulating RBCs indicated that many of them were lodged in the ducts of Cuvier, while the thrombocytes were mostly located in the caudal hematopoietic tissue (CHT). It is unclear why these cells never entered circulation; it did not appear to be a developmental delay, as the cells were still not circulating when examined at 5dpf. Homing is not completely defective in *son* morphants, however. Even though we observed decreased numbers of neutrophils in *mpx*:GFP fish, they still migrated throughout the embryo similar to uninjected controls. Additionally, *lck*:GFP T cells were decreased in morphants, but still were properly located in the thymus. Examination of *son's* role in homing or trafficking cells may be warranted in the future to see why these cells were specifically affected in this way.

It is important to note that we reduced the levels of Son in this study with a *son*-specific MO, which is a splice-blocking MO that significantly reduced the amount of full length *son*

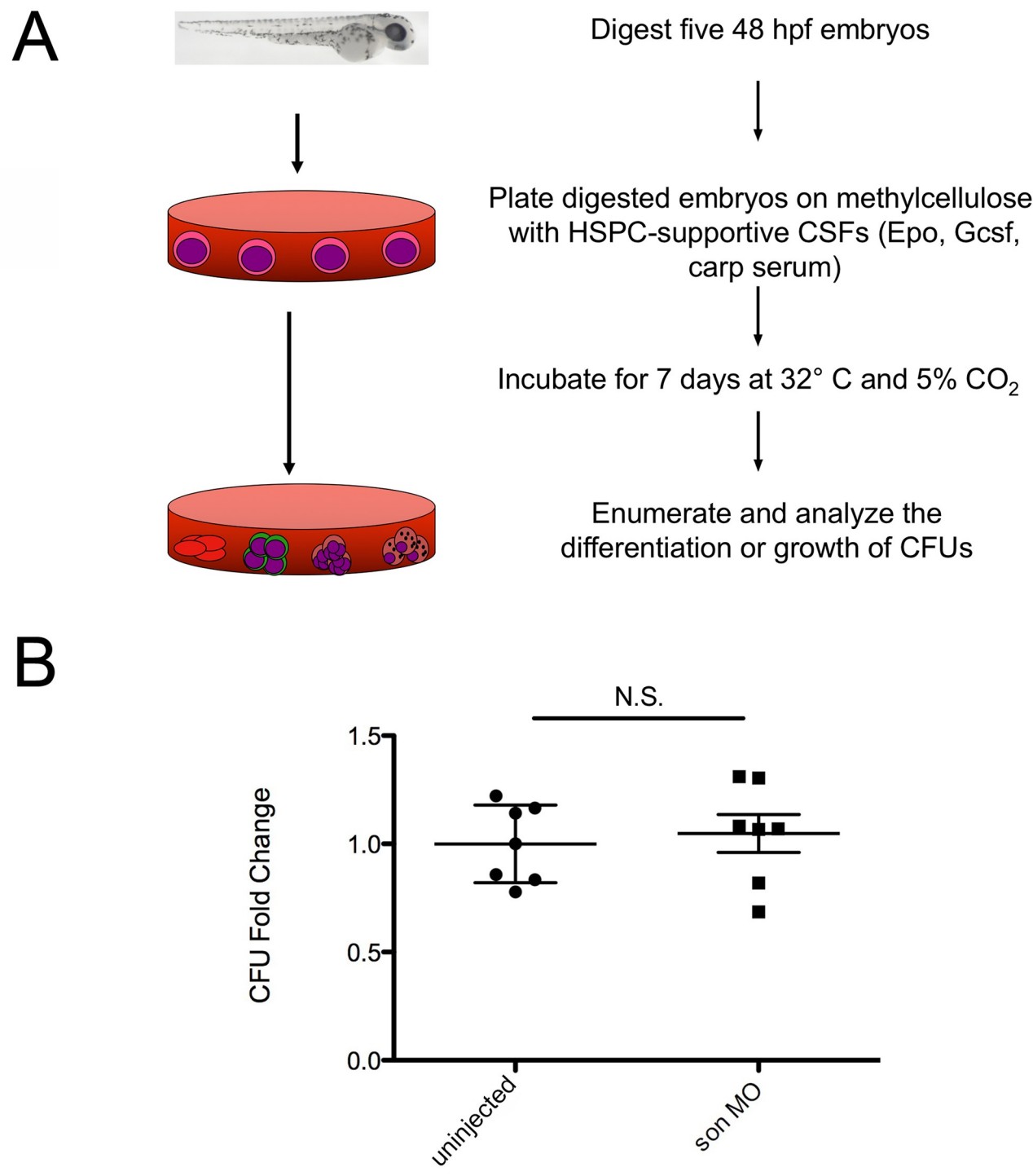

**Fig 5. *son* knockdown has little effect on erythromyeloid HSPC numbers.** (A) Experimental procedure. **(B)** wt embryos uninjected (left) or injected with *son* MO at the one-cell-stage of development (*son* MO, right) were dissociated at 48hpf and plated in methylcellulose media with Epo, Gcsf, and carp serum. Fold difference in colony forming units (CFUs) is shown. Each point represents ten embryos randomly selected from the respective conditions that were dissociated and plated. Middle lines represent mean and error bars represent SD. N.S. represents no significance.

mRNA present in the embryo. To examine the effect of *son* MO on blood cells, we examined the time at which they arose during development. Definitive RBCs and myeloid cells are first detected at 48hpf. Thrombocytes are first seen at 72hpf, and T cells arise at 5dpf (as reviewed in [46]). All images were taken and qRT-PCR was performed when mature blood cells were first seen; we did follow *son* morphants up to 5dpf, but we never saw a recovery in the number of blood cells. It is not useful to follow the fish for much longer than this, as MOs appear to start losing their effectiveness as early as 4dpf [47]. While it is difficult to say whether or not fish would recover normal blood cell counts after this time period, these data indicate that the blood reduction observed is not just a delay in the development of the hematopoietic system caused by MO injection. Observation of embryos injected with control MOs showed no blood defects, confirming these findings.

All of the cells that we examined are derived from definitive HSPCs during development, meaning that they differentiate from upstream progenitor cells. While RBCs are present before 48hpf, they are mostly primitive RBCs that do not derive from a definitive precursor like an erythromyeloid progenitor (EMP) or HSC [48]. Thrombocytes observed at 72hpf and neutrophils imaged at 48hpf also are derived from either EMPs or HSCs [48]. T cells, on the other hand, are derived from the differentiation of HSCs; they are lymphoid cells that cannot be generated by EMPs [48]. To examine if HSPCs were affected, we used an *in vitro* methylcellulose assay that can quantitate differences between normal and morphant fish. While other experimental assays exist to study and measure HSPCs, they all have their limitations. Microscopy of labelled cells can indicate putative progenitors, but one must follow these cells over time to see where they go and what they become in the developing embryo, which is challenging and requires specialized microscopy equipment. Transplantation experiments also exist, but are costly and time consuming. Transplantation is also usually used to prove long-term engraftment, so it is more useful for proving the existence of long-lived HSCs versus downstream short-lived HSPCs like CMPs, GMPs, and CLPs. In essence, methylcellulose assays are an alternative to quickly and efficiently determine if there is an imbalance in the number of HSPCs present in an organism. However, methylcellulose assays also have deficiencies; namely that they depend on the cytokines added to the medium for encouraging what cells will survive and proliferate. Additionally, you cannot detect T cells, which require a thymus to develop.

In our studies, the numbers of HSPCs grown with Epo and Gcsf were similar between the morphant and uninjected fish. This indicates that erythroid and myeloid differentiation is similar in the morphant animals when compared to uninjected animals. Importantly, human patients with *SON* deficiency have recurrent infections, low immunoglobulin levels, and clotting issues [18], suggesting that these patients have issues with hematopoiesis progressing properly from HSCs to definitive progenitors to mature blood cells. Human *SON* is located on human chromosome 21 and triplicated in Down syndrome, and these patients have elevated rates of AMKL, TMD, and other leukemias primarily caused by defects in the differentiation of HSPCs. Clearly further studies are warranted, possibly in another model system; CMPs, GMPs, and CLPs are speculated to exist in the zebrafish, but no transgenics or good assays exist to examine these cell types. There should also be attention given to blood cell types that were not addressed in our study, such as B cells, NK cells, dendritic cells, and basophils. All of these cells exist in zebrafish but are difficult to assay. Finally, it could be that *son* has an effect on the actual formation of the hematopoietic system, which derives from similar progenitors responsible for vasculogenesis. While the vasculature marker *fli1* appears to be completely unaffected in morphants, there may be some interest in understanding the biology of this gene regulation in the cardiovascular system, especially on the role it plays in the formation of blood. Previously we showed that *son* plays a role in kidney development [19]. As the main site of hematopoiesis in teleosts, it will be of interest to see if this delay in kidney development is

involved in aberrant hematopoietic development. Overall, these studies indicate that there should be further examination of the mechanism by which *son* affects HSPC formation and differentiation.

We have now shown that *son* affects brain [17], spine [17], kidney [19], and blood development in zebrafish. It is particularly interesting that both brain and blood development are affected by *son* since the brain develops from ectoderm while blood develops from mesoderm. One hypothesis for how one gene can affect both endoderm and mesoderm derived tissues is through niche signaling from the ectoderm to the mesoderm during HSPC development. Research suggests that ectoderm-derived neural crest cells contribute to the HSC niche, impacting HSC development [49]. Trunk neural crest cells, which become sympathetic nervous system neurons, physically associate with the dorsal aorta, the location of primary blood development, prior to hematopoietic initiation. Any disruption of this interaction results in impaired hematopoiesis [49]. Further examination of this is warranted- although *son* reduction does not reduce the number of HSPCs detected in our experiments, it may reduce the number of HSPCs that mature properly to give rise to mature blood. It may also play a role in their release from the niche to seed hematopoietic organs later in development or influence skewing of HSCs into CMPs, CLPs, or other downstream HSPCs. Another possibility is that *son* is regulating brain, spinal, and blood development through alternative mRNA splicing in these different tissues. *son* may operate pleiotropically, regulating distinct genes in different tissue types or by regulating different genes at different developmental time points. Differential gene regulation in this way could explain how *SON* is able to impact a variety of cell types. A closer look into the specific genes that *son* regulates may shed light on how one gene can affect the formation and homeostasis of many diverse tissues.

It is worth noting that zebrafish Son and human SON are not exactly the same, which may limit the use of zebrafish to understand certain human phenotypes related to specific mutations in SON, especially in a repetitive amino acid region that is lacking in the zebrafish ortholog. Son and SON share only 47% sequence identity, but do have a sequence identity of 48% in the RS domain (Ser/Arg-rich domain), 79% in the G-patch domain (an RNA binding domain), and 67% in the DSRM (double stranded RNA-binding motif). Importantly, the total homology between the proteins from the RS domain to the C terminus is 59% [17]. While these domains are essential for the function of Son and SON, not all human patients had mutations in these regions. In fact, 13/27 patients identified in our previous studies had mutations in a repetitive amino acid region upstream of the RS domain [17, 19]. While the zebrafish loss-of-function phenotypes seem to recapitulate the brain [17], spine [17], kidney [19], and blood phenotypes seen in human patients, it will be difficult to study specific mutations located in the repeat regions with the zebrafish model.

Overall, these studies indicate that *son* is involved in normal blood cell differentiation in a vertebrate model system. Additional research on the role of *son* in mammals such as mice and humans will help gain a clearer understanding of how *son* affects blood development and homeostasis, and how its perturbation plays a role in blood diseases.

## Supporting information

**S1 Fig. *son* mRNA shows two distinct sizes.** Full length son mRNA produced by the mMachine kit has two sizes: the top band is mRNA produced from circular *son* plasmid, and the bottom band is from linearized *son* plasmid.
(TIF)

**S2 Fig. *son* MO reduces full length *son* transcript.** 10 random embryos either uninjected or injected with *son* MO at the one-cell-stage of development (*son* MO) collected at 24hpf (left),

48hpf (middle), and 72hpf (right) were subjected to RT-PCR for *son* (top) and *ef1α* (bottom) transcripts.
(TIF)

**S3 Fig. *son* MO reduces RBC numbers.** Flow cytometry of *lcr*:GFP+ embryos at 48hpf either injected with control MO (**A**), *son* MO (**B**), or MO-injected with *son* RNA (rescue) (**C**). *lcr*:GFP is shown along the x-axis; y-axis is the red fluorescence channel, examined at the same time to gate out auto-fluorescent and light-refractive cells in the developing embryo. Each plot represents ten embryos randomly selected from control MO, MO-injected (*son* MO), and MO-injected with *son* RNA (rescue) conditions that were enzymatically dissociated and analyzed with a flow cytometer. The numbers in the gates are the percentage of GFP⁺ RBCs present. **(D)** Data presented in graphical format. Middle lines represent mean and error bars represent SD. * represents p = 0.05, ** represents p < 0.001, N.S. represents no significance.
(TIFF)

**S4 Fig. *son* knockdown does not negatively affect vasculature formation.** Representative images of 48hpf *fli1*:GFP embryos injected with control MO (top) or *son* MO (bottom) at the one-cell-stage of development show no significant differences in vessel formation or integrity associated with *son* MO injection. Brightfield images are shown in top left (40x), and fluorescent images are shown in top right (40x). Zoomed in fluorescent images are shown below (80x). Red arrows and brackets indicate the dorsal aorta, the site of HSC formation.
(TIF)

**S1 Data.**
(XLSX)

## Acknowledgments

We thank Betsey Tamietti for excellent laboratory management and Kathy Johns for administrative assistance.

## Author Contributions

**Conceptualization:** Rebecca L. Belmonte, Eun-Young Erin Ahn, David L. Stachura.

**Funding acquisition:** David L. Stachura.

**Investigation:** Rebecca L. Belmonte, Isabella L. Engbretson, Jung-Hyun Kim, Illiana Cajias, David L. Stachura.

**Resources:** Eun-Young Erin Ahn, David L. Stachura.

**Supervision:** Eun-Young Erin Ahn, David L. Stachura.

**Writing – original draft:** Rebecca L. Belmonte.

**Writing – review & editing:** Eun-Young Erin Ahn, David L. Stachura.

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
