## [Decision Letter · Decision Letter 0]

9 Sep 2020

PONE-D-20-22217

son is necessary for proper vertebrate blood development

PLOS ONE

Dear Dr. Stachura,

Thank you for submitting your manuscript to PLOS ONE. After careful consideration, we feel that it has merit but does not fully meet PLOS ONE’s publication criteria as it currently stands. Therefore, we invite you to submit a revised version of the manuscript that addresses the points raised during the review process.

We look forward to receiving your revised manuscript.

Kind regards,

Daniel R. Larson, PhD

Academic Editor

PLOS ONE

Journal Requirements:

2. In your Methods section, please include a comment about the state of the animals following this research. Were they euthanized or housed for use in further research? If any animals were sacrificed by the authors, please include the method of euthanasia and describe any efforts that were undertaken to reduce animal suffering.

3. Thank you for icluding your fundinng statement; "D.L.S. is a scientific consultant and has received compensation from Finless Foods, Inc. and Xytogen Biotech, Inc. This does not alter our adherence to PLOS ONE policies on sharing data and materials. All other authors declare no competing interests."

In your cover letter please confirm the aove as being your competing interests statement and we will amend this o your behalf

4. With regards to the above funding statement which is more appropraite as your competing interests statement;

Reviewers' comments:

Reviewer's Responses to Questions

**Comments to the Author**

1. Is the manuscript technically sound, and do the data support the conclusions?

Reviewer #1: Yes

Reviewer #2: Partly

2. Has the statistical analysis been performed appropriately and rigorously? 

Reviewer #1: Yes

Reviewer #2: Yes

3. Have the authors made all data underlying the findings in their manuscript fully available?

Reviewer #1: Yes

Reviewer #2: Yes

4. Is the manuscript presented in an intelligible fashion and written in standard English?

Reviewer #1: Yes

Reviewer #2: Yes

5. Review Comments to the Author

Reviewer #1: Criteria for Publication:

The study meets all criteria for publication. #3 criteria that “experiments, statistics, and other analyses are performed to a high standard and are described in sufficient detail” would benefit from minor revisions as outlined below to improve study robustness and clarity of methods utilized.

General Comments:

This is an interesting study investigating the effects of son reduction on different types of blood cells. It is especially interesting that son reduction does not result in a decrease in erythromyeloid HSPC numbers but does impair erythropoiesis through reduction of mature blood cells. Human SON LoF mutations result in spinal and brain malformations, and the patients also have hematologic and kidney abnormalities. This study uses zebrafish as a model organism to assess the SON ortholog role in regulating blood development. Overall, this manuscript has appropriate conclusions and demonstrates a clear association between son reduction and a decrease in multiple mature blood cell types. I especially appreciated the use of rescue experiments to show that son recovery is associated with an increase in blood cells close to normal levels. The discussion section is especially great and thoroughly explains the results, limitations, future work, and long-term hypotheses.

Conceptually, I am interested in how son’s lack of repetitive sequence motifs could affect the translation of these results to mammalian studies. As seen in Kim et al. 2016, several patients exhibited SON LoF mutations in the repeat domain, which does not appear to be conserved in zebrafish son. Although that same paper shows son depletion exhibits similar head and spinal malformations to SON depletion, it is still worth mentioning how the absence of the repeat motifs in son could limit this study.

Major points:

• This study would benefit from more detailed descriptions of methods and results. For example, how exactly were images captured/processed, and how was pixel density determined in ImageJ?

• Determining a threshold for each phenotype (no blood, no circulation, normal blood, little blood) would greatly increase the robustness of the experiments. How do you define these phenotypes; by eye or with quantitative measurements of blood cells? How can you reliably quantitate no circulation? An example is in Figure 3 where, at least to my untrained eye, the no circulation image appears very similar to the normal blood phenotype. Having a quantitative threshold to describe these phenotypes would take away any guesswork on which category each fish fell into.

Minor points:

• To improve clarity, figure 1&2, and 3&4 can be combined as they are describing the same cell type.

• Figure 2, p=0.04 should be * and p=0.02 should be **

• Could you provide exact significance values in your legends? I would also suggest keeping *,**,***,**** values consistent throughout the figures for clarity.

• Since you’re using uninjected fish as your control, figures should say “uninjected control” or similar instead of “control MO” to describe the condition because this implies control MO was injected.

• A brief explanation of why there is such a high T cell background in the yolk sac would be helpful.

• Figure S1: what are the fractions in the bottom right of the figure; fraction of fish displaying the phenotype? Also, explain the difference in son MO fish (since you included two different fish for this condition).

• Figure 7: A brief description in the results section of why you did this experiment would help. The description provided in the discussion is great but could be mentioned in an earlier section.

Reviewer #2: In this manuscript, David Stachura and colleagues, expand on their previous work to show that Son is required for efficient blood development in zebrafish embryo. The authors used a single, previously established morpholino to deplete Son expression at the single-cell stage of embryonic development and show reduced levels of differentiated blood cells using a panel of transgenic models. Importantly, the observed phenotypes were largely rescued following injection of son mRNA suggesting that the effect of the son morpholino were specific.

Overall, this is a nicely written manuscript that provides indication about an important role of Son in blood development. However, some control experiments that would further solidify the conclusion of the study should be included prior publication.

1. Although the Son morpholino used in this study has been used in a previous study, its efficacy and specificity has to be carefully examined. The authors mention on page 11 that the morpholino function by altering son splicing and reducing protein expression levels (presumably by eliciting non-sense mediated decay). In any case, the impact of the morpholino in splicing should be confirmed by RT-PCR. In addition, an RNA gel to show the integrity of the mRNA used for the rescue experiments should be shown.

2. The authors claim that there are no blood differences between uninjected embryos or those injected with control morpholinos (page 16). This is not supported by the referenced figure (Figure S1) as only representative images are shown and only one transgenic model is assessed. More images and models need to be compared. Ideally, in the future the authors could use control morpholinos to eliminate the need for comparison between control- and non-injected animals.

3. The study would be strengthened if the authors in addition to the imaging data also included flow cytometry analysis for the quantification of blood cell number. Perhaps in such an experiment, non-targeting morpholinos could be used as control to accommodate the previous point.

6. PLOS authors have the option to publish the peer review history of their article (what does this mean?). If published, this will include your full peer review and any attached files.

Reviewer #1: No

Reviewer #2: No

---

## [Author Response · Author response to Decision Letter 0]

19 Jan 2021

Please see document attached to manuscript.

Thanks to the editors and reviewers. We have responded to reviewer comments, and all our responses are below in red font. We believe that the manuscript is greatly improved and now suitable for publication.

We have made sure our manuscript and files meet PLOS ONE conventions.

2. In your Methods section, please include a comment about the state of the animals following this research. Were they euthanized or housed for use in further research? If any animals were sacrificed by the authors, please include the method of euthanasia and describe any efforts that were undertaken to reduce animal suffering.

We have added the following statement to the Methods section: “After experiments were performed all animals were returned to the aquarium system to be used for further research.” 

3. Thank you for including your funding statement; "D.L.S. is a scientific consultant and has received compensation from Finless Foods, Inc. and Xytogen Biotech, Inc. This does not alter our adherence to PLOS ONE policies on sharing data and materials. All other authors declare no competing interests."

In your cover letter please confirm the above as being your competing interests statement and we will amend this on your behalf.

We apologize for listing this as a “funding” statement; we agree that is more appropriately a “competing interests” statement. We have confirmed this in the revised cover letter.

4. With regards to the above funding statement which is more appropriate as your competing interests statement;

a. Please clarify the sources of funding (financial or material support) for your study. List the grants or organizations that supported your study, including funding received from your institution.

d. If you did not receive any funding for this study, please state: “The authors received no specific funding for this work.”

We have amended our “funding” statement to read: “This research was supported by the CSU Program for Education and Research in Biotechnology (CSUPERB) (New Investigator Awards to D.L.S.), NSF MRI (proposal 1626406), a California State University Chico Internal Research Grant (to D.L.S.), Student Research and Creativity Awards from California State University, Chico (to R.B.) and the NIH (R01CA190688 to E.-Y.E.A. and R15DK114732-01 to D.L.S.). The funders had no role in study design, data collection and analysis, decision to publish, or preparation of the manuscript.

We have added this to the revised cover letter.

Reviewers' comments:

Reviewer's Responses to Questions

Comments to the Author

1. Is the manuscript technically sound, and do the data support the conclusions?

Reviewer #1: Yes

Reviewer #2: Partly

We believe that we have addressed Reviewer #2’s concerns in the revised manuscript.

2. Has the statistical analysis been performed appropriately and rigorously? 

Reviewer #1: Yes

Reviewer #2: Yes

3. Have the authors made all data underlying the findings in their manuscript fully available?

Reviewer #1: Yes

Reviewer #2: Yes

4. Is the manuscript presented in an intelligible fashion and written in standard English?

Reviewer #1: Yes

Reviewer #2: Yes

5. Review Comments to the Author

Reviewer #1: Criteria for Publication:

The study meets all criteria for publication. #3 criteria that “experiments, statistics, and other analyses are performed to a high standard and are described in sufficient detail” would benefit from minor revisions as outlined below to improve study robustness and clarity of methods utilized.

We thank the reviewer for their constructive feedback and have made significant changes to the manuscript to address these issues. Hopefully the reviewer agrees that the revised manuscript is greatly improved and suitable for publication.

General Comments:

This is an interesting study investigating the effects of son reduction on different types of blood cells. It is especially interesting that son reduction does not result in a decrease in erythromyeloid HSPC numbers but does impair erythropoiesis through reduction of mature blood cells. Human SON LoF mutations result in spinal and brain malformations, and the patients also have hematologic and kidney abnormalities. This study uses zebrafish as a model organism to assess the SON ortholog role in regulating blood development. Overall, this manuscript has appropriate conclusions and demonstrates a clear association between son reduction and a decrease in multiple mature blood cell types. I especially appreciated the use of rescue experiments to show that son recovery is associated with an increase in blood cells close to normal levels. The discussion section is especially great and thoroughly explains the results, limitations, future work, and long-term hypotheses.

Conceptually, I am interested in how son’s lack of repetitive sequence motifs could affect the translation of these results to mammalian studies. As seen in Kim et al. 2016, several patients exhibited SON LoF mutations in the repeat domain, which does not appear to be conserved in zebrafish son. Although that same paper shows son depletion exhibits similar head and spinal malformations to SON depletion, it is still worth mentioning how the absence of the repeat motifs in son could limit this study.

We agree that further discussion of this is warranted, and have added the following to the discussion section of the manuscript:

“It is worth noting that zebrafish Son and human SON are not exactly the same, which may limit the use of zebrafish to understand certain human phenotypes related to specific mutations in SON, especially in a repetitive amino acid region that is lacking in the zebrafish ortholog. Son and SON share only 47% sequence identity, but do have a sequence identity of 48% in the RS domain (Ser/Arg-rich domain), 79% in the G-patch domain (an RNA binding domain), and 67% in the DSRM (double stranded RNA-binding motif). Importantly, the total homology between the proteins from the RS domain to the C terminus is 59%[17]. While these domains are essential for the function of Son and SON, not all human patients had mutations in these regions. In fact, 13/27 patients identified in our previous studies had mutations in a repetitive amino acid region upstream of the RS domain[17,19]. While the zebrafish loss-of-function phenotypes seem to recapitulate the brain[17], spine[17], kidney[19], and blood phenotypes seen in human patients, it will be difficult to study specific mutations located in the repeat regions with the zebrafish model.” 

Major points:

• This study would benefit from more detailed descriptions of methods and results. For example, how exactly were images captured/processed, and how was pixel density determined in ImageJ?

We apologize; we were trying to be brief in respects to methods. However, we realize that this may be confusing. We have added back information to these sections. Hopefully the reviewer agrees that there is now sufficient information to understand the methods and results.

• Determining a threshold for each phenotype (no blood, no circulation, normal blood, little blood) would greatly increase the robustness of the experiments. How do you define these phenotypes; by eye or with quantitative measurements of blood cells? How can you reliably quantitate no circulation? An example is in Figure 3 where, at least to my untrained eye, the no circulation image appears very similar to the normal blood phenotype. Having a quantitative threshold to describe these phenotypes would take away any guesswork on which category each fish fell into.

We appreciate this question, and should have better addressed this issue in the manuscript. First, we define these phenotypes by eye. The person looking at the animals looks at uninjected control animals first, and counts the amount of fish that are “normal.” Usually this is close to 100%. Then, they look at the morpholino-injected animals. It is usually quite striking, as in Figure 1, to see less blood versus normal, especially with RBCs. While the “normal” animals basically have their vessels completely full of green fluorescent cells (the vessels almost look like a constant “line” of cells), the “less blood” have scattered amounts of green cells moving through the vasculature. Honestly, it’s really easy to see visually- this isn’t just a small reduction in blood cells. Secondly, the “no circulation” phenotype, while difficult to observe in a still image, is very obvious under a microscope. Basically, you can see the heart pumping and fluid moving through the vessels, but the fluorescent cells are not moving at all. In other words, it looks like just plasma is circulating, with no blood cells. Your question about Figure 3: yes. It looks the same in a static picture. But in the “no circulation” animals those cd41+ cells are trapped in the caudal hematopoietic tissue- you never see them moving around in the animal’s vasculature. We aren’t sure if this is a homing issue for the cells or if the cells can’t extravasate properly, which we discuss in the Discussion. We have quantitated the amount of cells for the mpx:GFP animals by actually counting the myeloid cells. And lck:GFP animals were quantitated by pixel density. While there aren’t enough cd41+ cells at 72hpf to quantitate well with flow cytometry, we have added data showing flow cytometry experiments for RBCs. 

Minor points:

• To improve clarity, figure 1&2, and 3&4 can be combined as they are describing the same cell type.

We only separated them because we thought the images were easier to see if not condensed down into a smaller figure. We have put them together for clarity.

• Figure 2, p=0.04 should be * and p=0.02 should be **

We apologize for this oversight. It has been corrected in the revised manuscript.

• Could you provide exact significance values in your legends? I would also suggest keeping *,**,***,**** values consistent throughout the figures for clarity.

We have followed PLOS ONE conventions for this, which state:

“P-values. Report exact p-values for all values greater than or equal to 0.001. P-values less than 0.001 may be expressed as p < 0.001, or as exponentials in studies of genetic associations.”

Unfortunately, this policy doesn’t work for keeping *,**,***,**** values consistent throughout the manuscript. 

• Since you’re using uninjected fish as your control, figures should say “uninjected control” or similar instead of “control MO” to describe the condition because this implies control MO was injected.

We have corrected this throughout the manuscript.

• A brief explanation of why there is such a high T cell background in the yolk sac would be helpful.

We have added a brief discussion to the figure legend addressing this.

• Figure S1: what are the fractions in the bottom right of the figure; fraction of fish displaying the phenotype? Also, explain the difference in son MO fish (since you included two different fish for this condition). 

We agree that this was likely confusing. We have amended the figure (removing the fractions), and explaining it better in the Results section of the revised manuscript.

• Figure 7: A brief description in the results section of why you did this experiment would help. The description provided in the discussion is great but could be mentioned in an earlier section.

We have added a description of this to the Results section.

Reviewer #2: In this manuscript, David Stachura and colleagues, expand on their previous work to show that Son is required for efficient blood development in zebrafish embryo. The authors used a single, previously established morpholino to deplete Son expression at the single-cell stage of embryonic development and show reduced levels of differentiated blood cells using a panel of transgenic models. Importantly, the observed phenotypes were largely rescued following injection of son mRNA suggesting that the effect of the son morpholino were specific.

Overall, this is a nicely written manuscript that provides indication about an important role of Son in blood development. However, some control experiments that would further solidify the conclusion of the study should be included prior publication.

1. Although the Son morpholino used in this study has been used in a previous study, its efficacy and specificity has to be carefully examined. The authors mention on page 11 that the morpholino function by altering son splicing and reducing protein expression levels (presumably by eliciting non-sense mediated decay). In any case, the impact of the morpholino in splicing should be confirmed by RT-PCR. In addition, an RNA gel to show the integrity of the mRNA used for the rescue experiments should be shown.

We agree, and have added two supplemental figures that show these data.

2. The authors claim that there are no blood differences between uninjected embryos or those injected with control morpholinos (page 16). This is not supported by the referenced figure (Figure S1) as only representative images are shown and only one transgenic model is assessed. More images and models need to be compared. Ideally, in the future the authors could use control morpholinos to eliminate the need for comparison between control- and non-injected animals.

Honestly, we have never seen a difference between control-MO injected animals and uninjected animals in numerous other studies we have performed in the laboratory when we look at RBCs, thrombocytes, T cells, and myeloid cells. And, many other published reports indicate the same; that uninjected embryos closely recapitulate the phenotype seen with a control MO. So, to reduce cost and time, we just usually compare to uninjected animals or animals injected with PBS and phenol red. We have added more references to the manuscript to address this issue. And, we have added Supplemental Figure 3, showing that there is a significant reduction of cells in the morphant versus control MO injected groups. And, although we only show one image in Supplemental Figure 4, it was meant to represent 25 animals injected and observed. Hopefully the reviewer agrees that we show that control MO injected animals look like they develop normally, have normal vasculature, and recapitulate the uninjected vs. son-MO trends when it comes to RBC levels. 

3. The study would be strengthened if the authors in addition to the imaging data also included flow cytometry analysis for the quantification of blood cell number. Perhaps in such an experiment, non-targeting morpholinos could be used as control to accommodate the previous point.

We agree, and have added Supplemental Figure 3.

6. PLOS authors have the option to publish the peer review history of their article (what does this mean?). If published, this will include your full peer review and any attached files.

Do you want your identity to be public for this peer review? For information about this choice, including consent withdrawal, please see our Privacy Policy.

Reviewer #1: No

Reviewer #2: No

 We have no problem with making the reviews and our responses available to others.

---

## [Editor Report · Decision Letter 1]

9 Feb 2021

son is necessary for proper vertebrate blood development

PONE-D-20-22217R1

Dear Dr. Stachura,

We’re pleased to inform you that your manuscript has been judged scientifically suitable for publication and will be formally accepted for publication once it meets all outstanding technical requirements.

Kind regards,

Daniel R. Larson

Academic Editor

PLOS ONE
---

## [Editor Report · Acceptance letter]

15 Feb 2021

PONE-D-20-22217R1 

*son* is necessary for proper vertebrate blood development 

Dear Dr. Stachura:

I'm pleased to inform you that your manuscript has been deemed suitable for publication in PLOS ONE. Congratulations! Your manuscript is now with our production department. 

Kind regards, 

on behalf of

Dr. Daniel R. Larson 

Academic Editor

PLOS ONE